# GNN2GNN: Graph Neural Networks to Generate Neural Networks

**Andrea Agiollo**[1,2]                    **Andrea Omicini**[1]

[1]Department of Computer Science and Engineering (DISI), ALMA MATER STUDIORUM—University of Bologna, Italy
[2]The Research Hub by Electrolux Professional S.p.A., Pordenone, Italy

## Abstract

The success of neural networks (NNs) is tightly linked with their architectural design—a complex problem by itself. We here introduce a novel framework leveraging *Graph Neural Networks to Generate Neural Networks* (GNN2GNN) where powerful NN architectures can be learned out of a set of available architecture-performance pairs. GNN2GNN relies on a three-way adversarial training of GNN, to optimise a generator model capable of producing predictions about powerful NN architectures. Unlike Neural Architecture Search (NAS) techniques proposing efficient searching algorithms over a set of NN architectures, GNN2GNN relies on learning NN architectural design criteria. GNN2GNN learns to propose NN architectures in a single step – i.e., training of the generator –, overcoming the recursive approach characterising NAS. Therefore, GNN2GNN avoids the expensive and inflexible search of efficient structures typical of NAS approaches. Extensive experiments over two state-of-the-art datasets prove the strength of our framework, showing that it can generate powerful architectures with high probability. Moreover, GNN2GNN outperforms possible counterparts for generating NN architectures, and shows flexibility against dataset quality degradation. Finally, GNN2GNN paves the way towards generalisation between datasets.

## 1 INTRODUCTION

Deep Learning (DL) techniques have recently seen an unstoppable rise in popularity: DL has changed the approach to most intelligence task, ranging from vision to text and audio processing. Among those techniques, neural networks (NNs) have become the most popular solution. NN perform-ance is tightly linked with the operations they leverage and how these are connected – their architecture – whose design has been shown to be as complex as much as it is relevant.

On the other hand, there is no trivial way to find out the best NN design for a specific task. *Neural Architecture Search* (NAS) techniques have recently emerged to tackle NN design issue [Elsken et al., 2019]: the underlying concept of NAS approaches is to efficiently search for the best NN architecture over a set of structures defined as a search space. Despite its success, NAS exhibits several drawbacks (see Section 6), as it relies on looking for the best architecture rather than learning architectural criteria for building the optimal NN. Moreover, NAS approaches are not flexible with respect to slight changes of the application scenario, require huge amount of resources to run, and are limited by their search space specifications. Also inspired by these limitations, in this work we present GNN2GNN, a novel tool leveraging graph neural networks to generate NN architectures.

GNN2GNN is a meta-learning framework exploiting Graph Neural Networks (GNNs) to learn generating efficient NN structures. GNNs are particular models proposed to tackle *graph*-processing tasks via convolution-equivalent operation over graphs [Wu et al., 2021]. A NN structure can be seen as a Directed Acyclic Graph (DAG) where nodes represent layers – implementing common operations like convolution, pooling, etc. – and edges represent how the output of one layer is fed to the following one. In this context, we propose a three-way adversarial learning setup to allow GNN to learn the features of an efficient NN structure and generate novel architectures. More in details, a generator GNN is trained to produce plausible architectures, while a discriminator GNN is optimised to distinguish between generated and real architectures. Finally, a valuer GNN aims at optimising the performance of the generated architectures. During training, the generator loss is defined as a mixture of the discriminator and valuer feedbacks, therefore aiming at enabling the learning of realistic – i.e., discriminator feedback – and powerful – i.e., valuer feedback – architectures.

*Accepted for the 38th Conference on Uncertainty in Artificial Intelligence* (UAI 2022).

While being embeddable into a broader NAS approach, GNN2GNN represents a powerful approach to propose NN architectures by itself. Indeed, differently from NAS techniques, which aim at efficiently searching NN architectures over a set of available ones, GNN2GNN aims at intrinsically learning architectural criteria from a set of available architecture-performance pairs. While NAS consider to recursively propose, evaluate and optimise a set of NN structures (see Figure 1 left), we here consider learning to propose architectures from a set of NNs in a single step—i.e., training of the generator GNN (see Figure 1 right). Once trained, GNN2GNN is capable of proposing multiple efficient NN architectures at once, rather than focusing solely on the local optimum obtained from the deployed search algorithm. Therefore, GNN2GNN significantly shifts the paradigm of the approach to the problem of NN architecture design, from relying on *searching* architectures to *learning* design criteria.

To summarize, the contributions that our work brings are the following:

- We present GNN2GNN, the first – up to our knowledge – framework leveraging GNN to generate powerful NN architectures, without relying on inefficient searching procedure.

- We show the effectiveness of our framework over two state-of-the-art datasets, highlighting its flexibility and generalization capability.

## 2   PRELIMINARIES ON GRAPH NEURAL NETWORKS

As the proposed framework relies on graph manipulation via GNNs, here we briefly introduce Graph Neural Networks, presenting their fundamental concepts. Graph Neural Networks (GNNs) have been proposed as an extension of traditional NNs to enable processing of non-rigidly structured data such as graphs. GNNs are mathematical models operating upon directed graphs, whose vertices (respectively, arcs) are labelled with vectors (or matrices, or tensors) of real numbers – $\mathbf{x}_v \in \mathbb{R}^d$ for vertex $v$, and $\mathbf{a}_{v,w} \in \mathbb{R}^c$ for the arc between vertex $v$ and $w$ –, each one carrying further numeric information about the corresponding vertex (resp., arc). GNNs rely on *graph convolution*, which represents the generalisation of a 2-dimensional convolution over graph-structured data. Graph convolution is defined over a single vertex $v$ and its neighbourhood $N(v)$, and relies on three successive phases:

*propagation* — the information $\mathbf{x}_{v'}$ belonging to each vertex $v' \in N(v)$ is weighted by the information $\mathbf{a}_{v,v'}$ belonging to the arc among $v$ and $v'$, then propagated to vertex $v$;

*aggregation* — the information propagated from each ver-

tex $v' \in N(v)$ to $v$ is aggregated via an aggregation function;

*transformation* — the aggregated information corresponding to vertex $v$ is transformed into a new embedding vector and assigned back to vertex $v$, as its new state $\mathbf{x}'_v$ .

The single convolution operation is applied in parallel to each vertex in $G$, updating the whole graph representation.

GNNs have proven to be successful in many tasks involving graph structured data. Most common applications concerns computational chemistry [Fung et al., 2021], social recommendations [Fan et al., 2019], computer vision [Wang et al., 2019], and many others [Hamilton et al., 2017, Yu et al., 2018]. However, a comprehensive review of GNNs and the underlying techniques is clearly out the scope of this paper: therefore, we refer interested readers to Wu et al. [2021], Zhou et al. [2020].

## 3   GNN2GNN

In this section we present our framework, namely GNN2GNN. GNN2GNN leverages G̲raph N̲eural N̲etworks to G̲enerate N̲eural N̲etworks. We first present briefly how NN architectures can be mapped into graph structures (Section 3.1). We then introduce the general pipeline for generating and processing NN architectures (Section 3.2), focusing specifically on its components.

### 3.1   NEURAL NETWORKS AS GRAPHS

NN architectures are uniquely defined by a set of layers $\mathcal{L}$, a set of operations $\mathcal{O}$ applied on layers, and a set $\mathcal{I}$ of interconnections between layers. Each layer $l_v \in \mathcal{L}$, with $v \in [0, |\mathcal{L}|]$, identifies a specific component of the NN architecture and is characterised by a specific operation $o_v \in \mathcal{O}$. Interconnections between layers, on the other hand, define how layers are linked to each other. An interconnection $i_{v,w} \in \mathcal{I}$ identifies that layers $l_v$ and $l_w$ are connected, and more specifically, it identifies that the output of the operation $o_v$ applied at layer $l_v$ is used as an input for the operation $o_w$ applied at layer $l_w$. It is important to notice that, thanks to the feedforwarding nature of standard NNs, there exists total ordering among the layers in $\mathcal{L}$ and interconnections can only exist between successive layers. Mathematically speaking, $\exists i_{v,w} \in \mathcal{I} \iff w > v$.

Following the above notations, NN architectures can be mapped easily into graph structures, specifically to Directed Acyclic Graphs (DAGs). Layers in $\mathcal{L}$ are mapped into a set of vertices $\mathcal{V}$ characterised by a set of features $\mathcal{X}$ representing layers operations ($\mathcal{O}$), while interconnections ($\mathcal{I}$) are mapped into a set of directed edges $\mathcal{E}$. Vertices – i.e., layers –, along with their features – i.e., operations –, are defined as vectors $\mathbf{x}_v \in \mathbb{R}^d$, where $v$ enumerates the

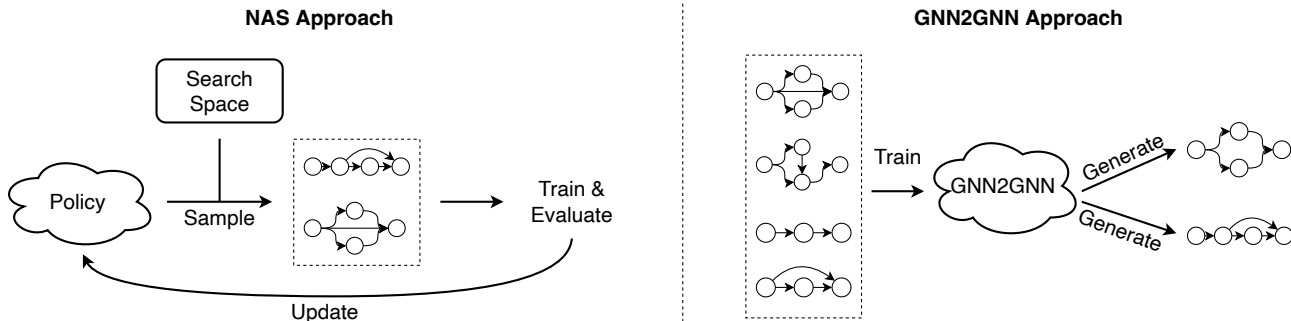

Figure 1: **Left**: NAS approaches rely on a recursive sampling, evaluation and optimisation procedure. A NAS policy is used to sample architectures from the search space. The sampled architectures are then trained and evaluated to optimise the NAS policy depending on their performance. Once a convergence criterion is met, NAS identifies the sub-optimal NN architecture. **Right**: The GNN2GNN approach rely on a single training procedure where GNN2GNN learns to propose effective NN architectures. The trained generator is able to produce multiple powerful NN architectures, rather than identifying solely the local sub-optimal NN architecture.

graph vertices, and $d$ represents operations cardinality. On the other hand, the set of edges – i.e., interconnections –, is denoted by the adjacency matrix $\mathbf{E} \in \mathbb{R}^{|\mathcal{V}| \times |\mathcal{V}|}$, where $\mathbf{e}_{v,w} = 1 \iff \exists i_{v,w}$. Therefore, a NN architecture can be mapped into a DAG defined by $\mathbf{X} \in \mathbb{R}^{|\mathcal{V}| \times d}$, where $row_k(\mathbf{X}) = \mathbf{x}_v$ – i.e. a matrix of vertices characterised by the operations they apply – and $\mathbf{E}$—i.e., the adjacency matrix defining how operations are connected.

## 3.2 ADVERSARIALLY GENERATE ARCHITECTURES

The proposed framework relies on a generative adversarial approach (GAN) [Goodfellow et al., 2014], applied over graph structured data leveraging Graph Neural Networks. The proposed framework is presented in Figure 2 and relies on three components:

- A *generator* GNN $G$ is in charge of proposing graph structures representing NN architectures.

- A *discriminator* model $D$ is responsible for distinguishing between NN structures proposed by $G$ and real architectures.

- A *valuer* network $V$ is responsible for predicting the architecture performance, therefore optimising the generation towards powerful structures.

GNN2GNN relies on such triplets of GNNs to allow $G$ to intrinsically learn optimal architectural criteria. Indeed, the discriminator and valuer models are used during training to optimise the generator status. More in details, the generator loss is defined as a mixture of the discriminator and valuer feedbacks:

$$\mathcal{L}_G = \lambda \cdot \underbrace{\mathcal{F}_D}_{D \text{ feedback}} + (1 - \lambda) \cdot \underbrace{\mathcal{F}_V}_{V \text{ feedback}} \tag{1}$$

Here, $\lambda$ represents the balancing factor between the two feedbacks. Leveraging such mixture loss, $G$ is capable of proposing realistic – i.e., $D$ feedback – and powerful – i.e., $V$ feedback – architectures. Finally, once trained, the GNN2GNN framework exploits solely the generator component to propose significant NN architectures.

### 3.2.1 Generator

Generating graph structures that satisfy specific properties is complex and represents an open research issue [You et al., 2018, Li et al., 2018]. This task complexity is three-folded:

**Q1** *Generate realistic structures.* For a generated structure to result realistic, the generative framework should learn which nodes should be linked and which not.

**Q2** *Generate realistic nodes.* The generated graph should be characterised by nodes having realistic features.

**Q3** *Stopping criteria.* In the generating process, it is important to identify when the generated graph structure has reached its final structure, which is non-trivial.

To tackle the aforementioned problems and generate realistic NN architectures, we here propose a novel generative GNN. Indeed, GNNs are particularly suited for handling interconnections and node features, while they exhibit limitations on stopping criteria identification. However, given the nature of available NAS techniques and datasets, this GNNs limitation is not an issue. Indeed, available NAS techniques restrict their searching space, limiting the number of layers – vertices – that compose the NN architecture. Therefore, publicly available NAS datasets build on top of this rationale, fixing the number of NN layers.

Building on top of the same rationale, we here propose a generator model that receives as input a fully-connected DAG – i.e., where every node is connected with every other

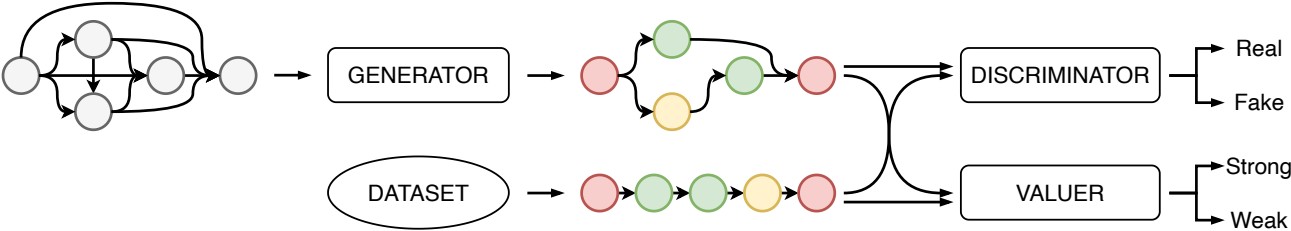

Figure 2: The GNN2GNN framework. The generator GNN produces NN architectures, starting from randomly initialised fully connected DAGs. The discriminator GNN aims at distinguishing artificial NN architectures from the real ones. The valuer network aims at predicting architectures performance, distinguishing between strong and weak structures or regressing their accuracy. Different colors of graph nodes represent different operations—embeddings. Red nodes identify input/output nodes, while green and yellow nodes may represent $3 \times 3$ convolution and max-pooling respectively. Gray nodes represent randomly initialised node embeddings.

node – having $N$ nodes. $N$ represents an hyperparameter of the framework, and can be arbitrarily set depending on the features of the NAS dataset at hand, or, on the complexity of the architecture to generate. Fixing $N$ immediately satisfies property **Q3**, implicitly setting a stopping criteria for the graph generation process. It is also important to notice that value $N$ only provides an upper limit on the number of layers composing the generated architectures. Indeed, architectures having $n \leq N$ can be generated by the proposed approach, thanks to edge removal and node isolation. Finally, node features of the input graph are randomly initialised, mirroring the usual GAN approach.

The proposed generator framework relies on four successive steps, presented in Figure 3 along with an example of input graph and generated architecture, and explained in details below.

**Graph convolution.** The generator applies $\mu$ layers of graph convolution to the randomly generated fully-connected graph received in input. Graph convolution layers allow elaboration of the random information received, building the backbone of the generated NN architecture. Depending on the number $\mu$ of convolutional layers selected, we should expect more or less fine-grained embeddings as output of this step. However, given the fully-connected nature of the input graph, a small value of $\mu$ is enough to obtain a meaningful graph embedding.

**Edge scoring & sampling.** Once a proposal of fully-connected architecture is obtained from the graph convolution layers, the generator applies a learnable scoring function to each edge of the graph at hand. This procedure allows different scores to be assigned to each edge of the architecture, depending on their relevance. To score edges we first build edge features vectors, through the concatenation of adjacent vertices features. Mathematically speaking, the feature vector of edge connecting vertex $v$ to vertex $w$ is $\mathbf{e}_{v,w} = \mathbf{x}_w \| \mathbf{x}_w$, where $\|$ denotes concatenation. Once the edge feature is obtained, the edge relevance is scored using a

standard densely-connected layer followed by normalisation in $[0, 1]$, obtaining $\mathbf{e}'_{v,w}$ which represents the score given to the edge between $v$ and $w$. To avoid non-differentiability issues that may arise from scores thresholding, edges are then sampled depending on their scores using a gumbel soft-max layer. This procedure ensures the survival of relevant – from the generator perspective – edges only, aiming at satisfying **Q1**. Edge scoring and sampling are here presented as a unique step, given their logical bond. However, it is also possible to conceive these two as separate steps, as done in Figure 3 to ease reader understanding of the framework.

**Layers operations generation.** The aim of this step is to assign one operation to each vertex – i.e., layer – of the graph corresponding to the NN architecture. To do so, the graph embedding obtained from the graph convolution step is combined with the sampled edge scores and used as input for a new layer of graph convolution. A softmax operation is then applied to the output of the convolutional layer to produce the one-hot encoding of the operations corresponding to each node. This specific step, aiming at identifying realistic nodes features – i.e., operations –, is meant to satisfy **Q2**. Here, the layer generation step focus solely on the layer type – i.e., operation to deploy –, ignoring layers dimensioning issues. Indeed, we consider layers size to be automatically inferrable from the overall NN architecture, as stated in Ying et al. [2019].

**Graph refinement.** Finally, the generator removes un-sampled edges from the graph as well as isolated nodes, obtaining the final NN architecture. Possible cycles and pending nodes are also removed during this step, ensuring therefore to produce a DAG architecture. The graph-refinement operation is left as the last operation to avoid possible non-differentiability issues which may arise from removal of nodes or edges. However, this does not influence the generation of layer operations, since zero-scored edges do not propagate information in the previous convolution step.

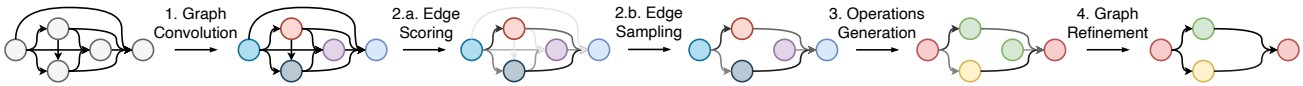

Figure 3: The generator receives in input randomly initialised – gray nodes – fully connected DAGs, and process them via graph convolution (1.). The new graph embedding, obtained from (1.) is used to score edges (2.a.). Light gray (dark gray) edges represent links having small (high) score. Edges are sampled (2.b.) and the scores are propagated to the next graph convolution step to obtain operations embedding (3.). Different colors of graph nodes represent different operations— embeddings. Red nodes identify input/output nodes, while green and yellow nodes may represent $3 \times 3$ convolution and max-pooling respectively. Finally, the graph is refined removing unsampled edges and nodes (4.).

### 3.2.2 Discriminator

The discriminator model aims at distinguishing between synthetically generated architectures and architectures available in the dataset at hand. In the proposed framework we build the discriminator model stacking $\nu$ layers of graph convolution, followed by a single densely-connected classification layer. Graph convolutional layers extract graph-structured features from the input graph, while the classification layer outputs a binary prediction. The complexity of the discriminator model – i.e., the number of graph convolutions $\nu$ – depends on the complexity of the architectures under inspection. Available NAS datasets consider fairly small architectures, as they deal with identical block structures, therefore in our experiments we set $\nu = 2$.

### 3.2.3 Valuer

The valuer model aims at identifying the performance of the architectures given as input. Structurally speaking, we build the valuer model similarly to the discriminator, stacking few layers of graph convolution, followed by a single densely-connected layer. The prediction of the valuer model over the structures generated by $G$ are also used for the generator optimisation, aiming to push $G$ toward the generation of more powerful NN architectures. Indeed, the generator model is trained minimising a combination between the standard GAN loss and the reward loss obtained from the valuer NN:

$$\mathcal{L}_G = \lambda \cdot \underbrace{\log(1 - D(G(z)))}_{\text{standard GAN loss}} + (1-\lambda) \cdot \underbrace{\mathcal{L}_R(V(G(z)))}_{\text{reward loss}} \quad (2)$$

where $z$ represents the randomly initialised graph used as input for $G$, $\mathcal{L}_R$ represents the reward loss and $\lambda$ represents a balancing factor between the two loss terms. The definition of $\mathcal{L}_R$ depends on the role of the final densely-connected layer of $V$, which can be used either to regress the performance of the graph at hand or to binary classify graphs—strong vs. not-strong architecture. In the first approach, $\mathcal{L}_R$ is represented via mean-squared error loss between the predicted performance of generated architecture and the best performing architecture. In the second, the reward loss is represented via cross-entropy loss between predicted classification and strong architecture labels. Our

experiments (see Section 4.4) show that the second approach is more consistent.

## 4 EXPERIMENTS AND RESULTS

In this section we propose a set of experiments to show the effectiveness of GNN2GNN for generating strong NNs. Our source code is available at https://github.com/AndAgio/GNN-2-GNN.

### 4.1 DATASETS

To test our framework performance we rely on the NAS101 [Ying et al., 2019] and NATS [Dong et al., 2021] benchmark datasets. Both datasets contain a set of NN architectures along with their recorded performance over a specific image classification task. Here, NN architectures are built from the repetition of identical cells, which are the target of our GNN2GNN approach. NAS101 contains $423k$ NN architectures trained multiple times over CIFAR-10 [Krizhevsky and Hinton, 2009]. On the other hand, NATS contains a set of $15k$ NN topologies trained over three different datasets: *(i)* CIFAR-10; *(ii)* CIFAR-100; *(iii)* ImageNet-16-120. However, NATS represent operations over graph edges, while GNN2GNN and NAS101 represent operations over graph nodes, as introduced in Section 3.1. Therefore, we translate NATS architectures into Section 3.1 form and remove possible duplicates, thus obtaining a refined version of NATS consisting of $7K$ unique architectures.

NAS101 and NATS datasets rely on similar search spaces used for the construction of NNs. Indeed, both consider a small set of operations, containing: *(i)* $3 \times 3$ convolution, *(ii)* $5 \times 5$ convolution, and *(iii)* pooling—NAS101 considers max-pooling, while NATS examines average-pooling. NAS101 contains NN cells with at most 7 nodes and 9 edges, while NATS examines cells with at most $8$ nodes, without imposing any restriction on the number of edges.

### 4.2 EXPERIMENTAL SETUP

To test GNN2GNN ability to produce novel architectures and generate strong cells, we remove part of the architec-

tures from the training dataset. We eliminate some randomly picked cells from the dataset, as well as the best 10% of architectures—w.r.t. their classification accuracy. Under these settings, the generator can not extract information from the strongest models during training, rendering the generation task more complex. Therefore, a generator capable of producing the best 10% of architectures is to be considered a strong model. $N = 10$ was selected since in NAS101 there exists quite a significant performance delta between the top-10% architectures and the rest.

Each GNN2GNN instance is trained for 20 epochs over the training set using standard Stochastic Gradient Descent and setting the learning rate to 0.001 and the batch size to 32. Moreover, during the first half of the training procedure we set $\lambda = 1$. This is done to allow $V$ properly learning to distinguish between strong and weak architectures, before leveraging it to optimise $G$ with backpropagation. Indeed, backpropagating information from a partially trained $V$ to the generator $G$ may increase the noise of its training, slowing down or hindering its optimisation. Therefore, in the first 10 epochs the generator model is optimised only through the discriminator $D$. After this setup period $\lambda$ is set back to its desired value, enabling the interaction between $G$ and $V$ as described by Equation (2).

## 4.3 EVALUATION METRICS

Throughout our experiments, we consider only models which always output valid NN architectures, since they output DAGs thanks to some refinement step. Therefore, the metrics that we define refer solely to the quality of the generated architectures. Moreover, since our framework is not directly comparable with NAS approaches, we avoid considering common NAS metrics—e.g., convergence time, etc. *Novelty* measures the percentage of generated architectures not belonging to the training set used. The *Top-N* metrics measure the percentage of generated architectures that belong to the best $N\%$ of architectures in terms of classification accuracy. $Acc_n$ measures the ratio between the number of generated architectures that reach an accuracy greater than $n$, and the number of generated models that belong to the dataset. Finally, $|Acc|$ measures the average accuracy reached by generated architectures.

## 4.4 ABLATION STUDY

To identify the best hyper-parameters setup for GNN2GNN, we propose a thorough ablation study. The ablation study is performed over the NAS101 dataset, given its higher degree of expressiveness w.r.t. NATS.

**Hyper-parameters** We consider the influence of the parametric values that may alter the generation of NN architectures. We take into account the balancing factor $\lambda$ used

during training, the temperature $\tau$ of the gumbel softmax layer used to perform edge sampling, and the number of graph convolution layers used by the generator $\mu$. Table 1 shows the results of the ablation study on such parameters. It is possible to notice that the model is highly affected by the balancing factor $\lambda$, which injects performance-critical information into the generator. Indeed, leveraging smaller $\lambda$ increases the performance of the proposed architectures, as the generator focuses more on the information received by $V$ through backpropagation. Smaller $\lambda$ values also improve GNN2GNN ability to predict more complex models. Architectures generated using $\lambda = 0.1$ have on average twice the number of parameters of their $\lambda = 1$ counterparts. This phenomenon is encouraging, as it shows that GNN2GNN is capable of mapping the whole space, thanks to $V$. However, smaller $\lambda$ increases the risk of mode collapse issues, as highlighted by the slight drop in novelty obtained with $\lambda = 0.01$. Finally, $\mu$ and $\tau$ do not seem to heavily influence the GNN2GNN performance.

Table 1: Ablation study over hyperparameters of $G$. Bold values highlight the best setup for each metric.

| Parameters | | | Novelty | Top-5 | Top-10 | Top-50 | $Acc_{90}$ | $|Acc|$ |
|---|---|---|---|---|---|---|---|---|
| $\mu$ | $\tau$ | $\lambda$ | | | | | | |
| 1 | 0.01 | 1 | 50.13% | 10.60% | 13.70% | 27.20% | 45.58% | 88.55% |
| | | 0.5 | 71.23% | 34.66% | 40.20% | 52.98% | 75.18% | 90.38% |
| | | 0.1 | 82.32% | 46.30% | 50.50% | 57.00% | 80.50% | 91.53% |
| | | 0.01 | 81.63% | 45.10% | 49.40% | 58.10% | 80.14% | 91.44% |
| | 0.1 | 1 | 51.79% | 12.10% | 15.40% | 25.20% | 40.23% | 88.10% |
| | | 0.5 | 67.81% | 19.01% | 22.62% | 39.20% | 59.23% | 89.48% |
| | | 0.1 | 82.52% | 45.19% | 50.53% | 58.91% | 80.60% | 91.48% |
| | | 0.01 | 82.47% | **46.50**% | 52.14% | **57.30**% | 79.32% | 91.35% |
| 2 | 0.01 | 1 | 51.66% | 8.57% | 11.40% | 26.83% | 41.63% | 88.53% |
| | | 0.5 | 73.84% | 40.41% | 45.28% | 54.61% | 75.01% | 90.89% |
| | | 0.1 | 82.06% | 46.09% | 51.03% | 57.82% | 81.55% | 91.54% |
| | | 0.01 | 82.23% | 45.66% | 50.20% | 57.19% | 79.69% | 91.42% |
| | 0.1 | 1 | 53.57% | 10.08% | 12.63% | 25.21% | 42.15% | 88.49% |
| | | 0.5 | 68.76% | 25.59% | 30.84% | 45.54% | 69.94% | 90.45% |
| | | 0.1 | **82.60**% | 45.91% | **52.21**% | 57.37% | **81.79**% | **92.04**% |
| | | 0.01 | 81.51% | 45.90% | 51.10% | 59.50% | 79.98% | 91.27% |

**Valuer mode** The mechanism used by the valuer network $V$ to identify strong and weak architectures may cause variation in the generation performance of GNN2GNN. We distinguish between a classification-based valuer C and a regression-based valuer R. The former identifies strong architectures as the cells belonging to the best half of the dataset. On the other hand, in the regression-based setup, $V$ aims at predicting precisely the classification accuracy of a cell from its structure. We pick the three best models in Table 1, retrain them using a regression-based $V$, and compare them against their classification-based counterparts.

Table 2 shows the results of the ablation study, highlighting the superiority of the classification-based setup. Indeed, regressing exactly NN performance from its architecture is complex, mostly since few small architectural modifications may lead to relevant performance changes. Such variability is complex to handle in a regression setup and hinders $V$ ability to predict correctly cells strength.

Table 2: Ablation study over evaluation mode adopted by $V$.

| Parameters | | | $V_{mode}$ | Novelty | Top-5 | Top-10 | Top-50 | $Acc_{90}$ | \|Acc\| |
|---|---|---|---|---|---|---|---|---|---|
| $\mu$ | $\tau$ | $\lambda$ | | | | | | | |
| 2 | 0.1 | 0.1 | C | **82.60%** | **45.91%** | **52.21%** | 57.37% | **81.79%** | **92.04%** |
| | | | R | 72.10% | 26.59% | 32.92% | 50.10% | 67.11% | 89.93% |
| 1 | 0.1 | 0.01 | C | 82.47% | 46.50% | 52.14% | 57.30% | 79.32% | 91.35% |
| | | | R | 71.06% | 25.90% | 34.13% | 51.07% | 65.79% | 90.07% |
| 2 | 0.1 | 0.01 | C | 81.51% | 45.90% | 51.10% | **59.50%** | 79.98% | 91.27% |
| | | | R | 70.33% | 27.04% | 33.54% | 50.97% | 66.43% | 90.01% |

In the remainder of the experiments, we build the GNN2GNN model employing a classification-based $V$ and the best hyperparameters values—i.e. $\mu = 2$, $\tau = 0.1$, $\lambda = 0.1$, as highlighted in Table 1. Indeed, these values represent a good starting point for deploying GNN2GNN over multiple scenarios, given NAS101 generality.

## 4.5 PERFORMANCE COMPARISON

To show the effectiveness of the proposed approach, we compare GNN2GNN against other generative mechanisms. We first consider generating random NN architectures using the Erdös–Rényi model [Erdös et al., 1960]. We then evaluate the strength of our approach against two GAN-based frameworks, relying on different generation strategies:

**MOLGAN-like** The model generates nodes and edges independently and simultaneously, recalling the approach by De Cao and Kipf [2018]. Two matrices representing node types and connections between them are generated from a random input vector and sampled using gumbel softmax.

**RNN** The model generates architectures starting from a single input node and appending new vertices – with corresponding edges – until a stopping criteria is met. This approach resembles the one by Zhang et al. [2019] and leverages Recurrent NNs to deal with graph construction via recursive node appending.

To make the comparison fair, both the MOLGAN-like and the RNN model are built using the three-way NNs adversarial approach that characterises GNN2GNN. Therefore, the three approaches differ solely on the generation criteria embodied by the generator model $G$.

Table 3 shows the performance of the different models. GNN2GNN vastly outperforms the counterparts, as it produces more accurate predictions for strong NN architectures. Moreover, more than $80\%$ of the predictions performed by our model are NNs characterised by an accuracy greater than $90\%$, while the best counterpart model – i.e., MOLGAN – fails to reach even $60\%$. This proves GNN2GNN's generation consistency. Indeed, even the simple random generation approach can sporadically generate powerful architectures, as shown also in Xie et al. [2019]. However, it suffers in terms of consistency, as it is uncommon to obtain articulate

architectures starting from a random empty graph.

Table 3: Performance comparison between GNN2GNN and other GAN based approaches to generate NN architectures.

| Dataset | Model | Novelty | Top-5 | Top-10 | Top-50 | $Acc_{90}$ | \|Acc\| |
|---|---|---|---|---|---|---|---|
| NAS101 | GNN2GNN | 82.60% | **45.91%** | **52.21%** | **57.37%** | **81.79%** | **92.04%** |
| | MOLGAN | 65.41% | 22.63% | 27.29% | 45.20% | 59.71% | 89.39% |
| | RNN | **96.34%** | 1.69% | 2.32% | 4.81% | 53.04% | 89.32% |
| | Random | 51.81% | 11.17% | 13.74% | 28.43% | 43.18% | 88.54% |

## 4.6 RESISTANCE TO DATASET QUALITY DEGRADATION

To study the flexibility of our approach against poorly-constructed datasets, we analyse GNN2GNN performance when a high number of strong models are removed from the training dataset. More in details, we first remove the best $N\%$ of models from the NAS101 training set, varying $N$ between 10 and 90, then retrain GNN2GNN. Table 4 shows these tests results. The performance loss between different setups is minimal, highlighting GNN2GNN strength against dataset quality degradation. Indeed, even when almost all best models are removed from the training set, GNN2GNN produces strong predictions, showing just a $3\%$ loss in the Top-5 metric and a $0.82\%$ decrease of the average accuracy reached by generated models.

Table 4: Performance comparison when the top $N\%$ of best architectures is removed from the training dataset.

| Dataset | $N$ | Novelty | Top-5 | Top-10 | Top-50 | $Acc_{90}$ | \|Acc\| |
|---|---|---|---|---|---|---|---|
| NAS101 | 10% | 82.60% | 45.91% | 52.21% | 57.37% | 81.79% | 92.04% |
| | 20% | 83.01% | 45.54% | 52.32% | 58.48% | 82.05% | 91.98% |
| | 30% | 83.67% | 46.10% | 51.07% | 57.04% | 80.14% | 91.68% |
| | 40% | 84.89% | 45.80% | 51.03% | 58.71% | 81.03% | 91.55% |
| | 50% | 85.00% | 44.01% | 48.81% | 56.30% | 79.24% | 91.33% |
| | 60% | 84.58% | 44.40% | 49.12% | 57.72% | 80.30% | 91.30% |
| | 70% | 84.20% | 44.66% | 49.38% | 56.71% | 79.52% | 91.37% |
| | 80% | 84.33% | 43.90% | 48.27% | 55.51% | 78.16% | 91.27% |
| | 90% | 85.71% | 42.77% | 46.70% | 55.72% | 78.24% | 91.22% |

## 4.7 GENERALISATION BETWEEN DATASETS

We now consider the generalisation ability of our framework. We start by training GNN2GNN over NATS and showing its performance. As Table 5 shows, the performance obtained over NATS are poor, probably due to the small size of NATS—i.e., only $7K$ NN architectures. We then apply the generator model trained over NAS101 to NATS, analysing its prediction performance. Table 5 shows the results of our generalisation study. While still not being satisfactory, we notice that performance strongly increase when GNN2GNN is transferred from NAS101 to NATS. This is encouraging, especially if we consider the strong difference between NAS101 and NATS. Indeed, only 576 NATS archi-

tectures are available also in NAS101, and their performance vary on average by $16.183\%$ between the two datasets.

Table 5: Performance of GNN2GNN when generalising between different datasets. Subscript refers to NATS split. $C10$ and $C100$ stand for CIFAR10 and CIFAR100, while $I$ stands for ImageNet.

| Dataset | | Novelty | Top-5 | Top-10 | Top-50 | \|Acc\| |
|---|---|---|---|---|---|---|
| Train | Test | | | | | |
| NATS$_{C10}$ | NATS$_{C10}$ | **76.73**% | 1.65% | 3.39% | 15.61% | 68.91% |
| NAS101 | NATS$_{C10}$ | 73.67% | **2.64**% | **5.08**% | **16.55**% | **70.25**% |
| NATS$_{C100}$ | NATS$_{C100}$ | 71.71% | 1.30% | 3.28% | 13.10% | 33.03% |
| NAS101 | NATS$_{C100}$ | **72.03**% | **2.64**% | **4.82**% | **16.90**% | **35.40**% |
| NATS$_I$ | NATS$_I$ | 81.03% | 0.91% | 2.17% | 8.42% | 16.75% |
| NAS101 | NATS$_I$ | **82.30**% | **1.93**% | **3.64**% | **11.49**% | **18.84**% |

## 4.8 PRELIMINARY COMPARISON AGAINST NAS

GNN2GNN does not represent a traditional NAS technique, as it does not rely on search space exploration and focuses solely on the architecture generation procedure. However, we can compare GNN2GNN against state-of-the-art NAS in terms of the performance obtained by the generated architectures over NAS101. Results shown in Table 6 are extracted from Yu et al. [2020] and consider 1000 GNN2GNN generation samples. The average accuracy of GNN2GNN generation is comparable with other NAS approaches. Meanwhile, results show that GNN2GNN vastly outperforms NAS techniques in terms of best accuracy. Indeed, the best architecture generated by GNN2GNN achieves $94.32\%$, while NAO tops up at $93.33\%$. Moreover, the architecture generated by GNN2GNN achieves a lower rank value, meaning that they are closer to the optimal architecture. Indeed, the best achievable accuracy in NAS101 is $95.06\%$, which represents an increase of only $0.72\%$ compared to what GNN2GNN achieves.

Table 6: GNN2GNN performance against state-of-the-art NAS approaches over NAS101. Subscript refers to the percentage of samples removed from NAS101.

| Model | \|Acc\| | Best Acc | Best Rank |
|---|---|---|---|
| DARTS Liu et al. [2019] | 92.21% | 93.02% | 57079 |
| NAO Luo et al. [2018] | **92.59**% | 93.33% | 19552 |
| ENAS Pham et al. [2018] | 91.83% | 92.54% | 96939 |
| GNN2GNN$_{10}$ | 92.04% | **94.32**% | **5372** |
| GNN2GNN$_{30}$ | 91.68% | 94.18% | 7843 |
| GNN2GNN$_{60}$ | 91.30% | 94.01% | 9371 |
| GNN2GNN$_{90}$ | 91.22% | 93.69% | 12570 |

## 5 DISCUSSION

GNN2GNN relies on an architecture-performance pairs dataset to remove part of the complexity burden of extracting architectures performance. This might represent a possible drawback, as it requires the training of a set of hand-crafted

NNs. However, results of Section 4.6 show how GNN2GNN learns to generate effective NNs even when most – i.e., 90% – of the arch-performance pairs are not available. Moreover, GNN2GNN does not impose any requirement on the dataset quality, as it can generate powerful architectures even when trained on the worst part of the dataset—i.e., worst 10%. Finally, Section 4.7 hints how GNN2GNN can translate the generation process to a new setup, without requiring the extraction of a new dataset.

## 6 RELATED WORK

The propose framework is tightly related to state-of-the-art techniques for searching NN structures—namely, Neural Architecture Search. NAS techniques have been recently proposed to tackle NN design [Elsken et al., 2019, Ren et al., 2021]. NAS techniques define a search space $\mathcal{S}$, containing NN architectures. A set of architectural rules identify the list of operations available at NN layers, as well as a list of rules defining admissible connections between NN layers. NAS approaches aims then at efficiently explore $\mathcal{S}$ to identify the strongest NN architecture. While the search space is explored, architectures are sampled, the corresponding NNs are built, trained over the task at hand, and evaluated, depending on a performance estimation strategy. Proposed NAS approaches may vary for the selected search space and the exploration strategy they deploy [Real et al., 2019, Tan and Le, 2019, Chu et al., 2020, Agiollo et al., 2021].

NAS approaches have proven to be successful in identifying powerful NN architectures. However, these approaches present drawbacks, which raise concerns about their applicability, namely:

*Learning to search vs. learning to build.* Whereas NAS approaches aim at efficiently searching a sub-optimal NN among the ones available in $\mathcal{S}$ – i.e., *learning to search* –, they fail to learn any proper architectural criteria — i.e., *learning to build*. In contrast, GNN2GNN, relying on graph learning techniques, aims at implicitly learning NN design criteria, representing a step forward towards *learning to build* NNs.

*Flexibility.* NAS techniques currently lack in flexibility and generality, as they cannot identify architectural criteria and focus on a specific task and dataset. Instead, GNN2GNN is limited only by the dataset at hand, being adaptable to the most diverse architectures, operations, and interconnections. Moreover, aiming to learn architectural criteria, GNN2GNN paves the way towards generalisation between datasets.

*Search space restrictions.* Most, if not all, NAS techniques limit the number of available operations or the way in which they can be connected to ease the searching procedure over $\mathcal{S}$. GNN2GNN exploits GNN, which are applicable to any graph structure, avoiding restrictions on architectural rules.

Our work is also related to the application of GNNs to NN architectures. Indeed, some works have exploited, with some success, the graph processing nature of GNNs to extract relevant information about NNs from their architecture. A common approach here consists of predicting the performance of a NN from its design, aiming to avoid expensive training processes [Lukasik et al., 2020]. Few of them even integrate GNNs into NAS algorithms [Yan et al., 2020, Wen et al., 2020]. Such frameworks, however, rely on the GNN regressive power solely to evaluate rapidly the performance of the proposed architectures, removing the training process from the NAS loop. Therefore, such approaches exploit GNN mostly as a speed-up component for the NAS procedure, failing to capture the proper expressive power of GNNs, which is their capability to learn and sub-symbolically express NN architectural criteria. Other works aim at finding relevant embeddings for the NN architectures at hand. Such embeddings identify a continuous embedding space, which can then be used to optimize the NN structure [Luo et al., 2018] or propose quick search mechanisms [Li et al., 2020].

# 7 CONCLUSIONS AND FUTURE WORKS

In this paper we present a novel GNN-based three-way adversarial framework for learning to generate strong NN architectures. The experiments completed over two state-of-the-art datasets highlight the strength of our approach. We show GNN2GNN ability to predict optimal NN architectures and its superiority against other available generation approaches. Moreover, given its flexibility against dataset quality degradation, the proposed framework represents a step forward towards learning architectural criteria for NNs design. Indeed, the GNN2GNN generator is capable of predicting unseen strong architectures even when dealing with unsound dataset. Finally, some experiments on knowledge transferability suggest the generalisability of our approach.

While aiming to overcome NAS limitations – via removal of search algorithms – GNN2GNN can also be integrated into a NAS approach as a proposal technique. Here, the adversarial framework characterising GNN2GNN would require online training, similarly to other NAS approaches. The implementation of a GNN2GNN-based NAS approach, along with thoroughly comparison with other NAS, is left for future works. Finally, we also intend to focus on boosting GNN2GNN generalisation ability, by introducing dataset embedding techniques.

### Acknowledgements

This paper has been partially supported by the CHIST-ERA IV project "EXPECTATION" (G.A. CHIST-ERA-19-XAI-005).

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
