# OpenReview forum: "GNN2GNN: Graph Neural Networks to Generate Neural Networks"
_auai.org/UAI/2022/Conference — UAI 2022 Poster_

### Official Review · Reviewer_2SkG · 2022-04-07

**Q2(1) Originality/Novelty:** 2
**Q2(2) Significance/Impact:** 2
**Q2(3) Correctness/Technical Quality:** 3
**Q2(6) Clarity Of Writing:** 3
**Q6 Overall Score:** 6
**Q8 Confidence In Your Score:** 4

**Q1 Summary And Contributions:**

The paper proposes a framework to train Graph neural networks in an adversarial fashion in order to generate neural network architectures. The major difference from NAS is that it learns the neural net architectures in a single step rather than the recursive approach of NAS. The idea is simple but quite intuitive and the experimental results show that using GNNs to generate neural networks is indeed an effective method.

**Q2 Assessment Of The Paper:**

More detailed information regarding each of these aspects is given below:

**Q2(4) Quality Of Experiments (Optional):**

2: Fair: The experimental evaluation is weak: important baselines are missing, or the results do not adequately support the main claims.

**Q2(5) Reproducibility:**

3: Good: Key resources (e.g., proofs, code, data) are available and key details (e.g., proofs, experimental setup) are sufficiently well-described for competent researchers to confidently reproduce the main results.

**Q3 Main Strengths:**

1. The paper is very well written and gives an innovative substitute to neural architecture search. The use of GNNs  to generate the neural net structure is clever as it has been known in literature that neural nets can be viewed as DAGs.

2. The biggest advantage of the paper lies in the simplicity of the approach. It clearly shows how and why is it different from NAS approaches.

3. The paper boasts of good ablation studies showing the robustness of the method.

**Q4 Main Weakness:**

1. Although simple, I can see several problems with this approach due to the adversarial training approach. The 1st one is stability of training. Its a well known fact that GAN-based methods have problems regarding stability while learning as it suffers from mode collapse. It can be a case here as well that same architecture is being learned every time. Can the authors comment if they faced this. If yes, then how was this handled and if no, why not?

2. Another problem I can see is the access to real architecture. What are these real architectures and how is it guaranteed that these are the best architectures for the underlying problem? Or the real architectures are ones that somewhat work?

3. The experiments are missing comparisons to NAS based methods. Since the paper motivates a lot with respect to NAS and how it is better, it would have been great to see some comparisons.

**Q5 Detailed Comments To The Authors:**

In addition to points in Q4, I have some more comments that I summarize here.

1. In section 3.2.1 what is meant by realistic features in Q2?

2. In the layers operation generation how does it specify a said operation to the node? It is not clear with the provided text.

3. I do not understand how the process of edge relevance defined in the paper actually represent relevance of an edge. There are several accepted  indices on the edge significance such as the edge betweenness centrality, degree product, bridgeness, diffusion importance, topological overlap and k-path edge centrality. How is a simple concatenation -> fully connected layer -> normalization leading to edge relevance should be described clearly in the paper .

Overall, although I appreciate the idea and the simplicity I think the paper needs more work before being ready for publication.

**Q7 Justification For Your Score:**

I have read the paper thoroughly and have also worked on a similar area. The points mentioned in Q4 were major considerations while providing the score since these are important questions that the paper should cover according to the reviewer.

**Q9 Complying With Reviewing Instructions:**

1: Yes.

---

### Official Review · Reviewer_ToFj · 2022-04-12

**Q2(1) Originality/Novelty:** 3
**Q2(2) Significance/Impact:** 3
**Q2(3) Correctness/Technical Quality:** 3
**Q2(6) Clarity Of Writing:** 3
**Q6 Overall Score:** 7
**Q8 Confidence In Your Score:** 4

**Q1 Summary And Contributions:**

To design better architecture of neural network, the paper proposes an adversarial training based GNN model called GNN2GNN to optimise a generator model capable of producing predictions about powerful NN architectures. Unlike the nowadays existing methods that only do searching algorithm on a given limited set of NN architecture, GNN2GNN is learning based and it can enjoy a variety. It also avoids the expensive and inflexible searching process like the current existing methods do.

**Q2 Assessment Of The Paper:**

More detailed information regarding each of these aspects is given below:

**Q2(4) Quality Of Experiments (Optional):**

3: Good: The experimental evaluation is adequate, and the results convincingly support the main claims.

**Q2(5) Reproducibility:**

3: Good: Key resources (e.g., proofs, code, data) are available and key details (e.g., proofs, experimental setup) are sufficiently well-described for competent researchers to confidently reproduce the main results.

**Q3 Main Strengths:**

1.	This paper firstly gives out an adversarial GNN model design to generate better and more realistic NN architectures, the idea is novel and ground-breaking, and it works theoretically.
2.	Since NN architectures have a solid impact on the success of neural network, thus, improving the NN architectures as well as the related topics are of importance and worth studying further for the method in this paper.
3.	This paper’s idea is technically sound, its basic theory and model design are clear and easy to understand and obey the intuition.
4.	The paper is of good reproducibility and the experiment results confirm its main claims.
5.	The paper is well-organized and clearly written, especially its figures to explain the model’s overview, easy to read and understand.


**Q4 Main Weakness:**

1.	The method generating NN architecture in this paper is not proved for its adaptability on different dataset, like unseen or even unsound ones.
2.	Time complexity for training this adversarial model is not clear, and how it is compared to the current state-of-the-art methods.
3.	The value of hyper-parameters in this paper is determined by given dataset, but when trained in other dataset, the selection of optimal hyper-parameters, especially how to balance the loss of both discriminator and valuer for generator (here the author adds a valuer in the model for three way learning) is of problem.
4.	The criteria of judging best NN architectures in this paper is designed by authors, and there is no other direct criteria that can be more fair to compare the method in the paper with the current state-of-the-art methods.


**Q5 Detailed Comments To The Authors:**

How is the time complexity of GNN2GNN, how is it compared to NAS?
Why not using a direct metric to judge the quality of NN architecture and compares GNN2GNN with NAS and other current methods based on that?
What’s the performance of GNN2GNN when it is on unseen or even unsound dataset? What’s the performance comparation then?


**Q7 Justification For Your Score:**

The novel design and its idea of the model (GNN2GNN) in this paper and excellent presentation weights most for the overall assessment, and theoretically the idea in this paper is of good practice and easy to use. And the main weakness are: training time issue, the selection of hyper-parameter as well as the generalization on other dataset are unknown and not mentioned in the paper.

**Q9 Complying With Reviewing Instructions:**

1: Yes.

---

### Official Review · Reviewer_dJoe · 2022-04-14

**Q2(1) Originality/Novelty:** 2
**Q2(2) Significance/Impact:** 2
**Q2(3) Correctness/Technical Quality:** 2
**Q2(6) Clarity Of Writing:** 3
**Q6 Overall Score:** 5
**Q8 Confidence In Your Score:** 3

**Q1 Summary And Contributions:**

This work proposes a graph neural network-based framework, GNN2GNN, for the generation of neural network architectures. The authors started with the motivation that the NAS approach based on search algorithms is expensive and inflexible. The proposed GNN2GNN can generate a set of NN architectures in one step. The experimental part is to validate the generation effect of GNN2GNN on two NAS datasets.

**Q2 Assessment Of The Paper:**

More detailed information regarding each of these aspects is given below:

**Q2(5) Reproducibility:**

3: Good: Key resources (e.g., proofs, code, data) are available and key details (e.g., proofs, experimental setup) are sufficiently well-described for competent researchers to confidently reproduce the main results.

**Q3 Main Strengths:**

S1: If the firstness of the work is proven, the novelty of the work can also be proven. However, research work exists to model neural network architecture search as a graph generation process. Of course, the novelty of GNN2GNN in that the generation process does not rely on the search algorithm still seems to be considered.

S2: The experiment can somewhat demonstrate the effect of GNN2GNN.

**Q4 Main Weakness:**

W1: Research work exists to model neural network architecture search as a graph generation process. For example, the work published at ICLR 2019, GRAPH HYPERNETWORKS FOR NEURAL ARCHITECTURE SEARCH.

W2: The work is missing some of the key NAS metrics, is it possible to compare them in a surrogate way?

**Q5 Detailed Comments To The Authors:**

Please add some related work, especially based on graph networks.

**Q7 Justification For Your Score:**

The novelty of this work needs to be better explained.

**Q9 Complying With Reviewing Instructions:**

1: Yes.

---

### Official Review · Reviewer_j5LS · 2022-04-16

**Q2(1) Originality/Novelty:** 3
**Q2(2) Significance/Impact:** 3
**Q2(3) Correctness/Technical Quality:** 3
**Q2(6) Clarity Of Writing:** 2
**Q6 Overall Score:** 4
**Q8 Confidence In Your Score:** 4

**Q1 Summary And Contributions:**

The author proposes to use GNN-based generative adversarial networks for learning neural network architecture design criteria. The model is based on a graph generator, a discriminator for learning to generate new architectures, and a “valuer” for neural architecture performance prediction. Experiments on NAS-bench101 and NATS are performed.

**Q2 Assessment Of The Paper:**

More detailed information regarding each of these aspects is given below:

**Q2(4) Quality Of Experiments (Optional):**

2: Fair: The experimental evaluation is weak: important baselines are missing, or the results do not adequately support the main claims.

**Q2(5) Reproducibility:**

2: Fair: Key resources (e.g., proofs, code, data) are unavailable but key details (e.g., proof sketches, experimental setup) are sufficiently well-described for an expert to confidently reproduce the main results.

**Q3 Main Strengths:**

1. Using GNNs as a graph generator in NAS is novel.

2. The proposed model is well explained and illustrated.

3. The proposed model is capable of generating well-performing neural architectures in the two datasets studied.


**Q4 Main Weakness:**

Please see my detailed comments.

**Q5 Detailed Comments To The Authors:**

1. The paper proposed to use a Generative Adversarial Network, and use a discriminator to distinguish “real” neural architectures from “generated” ones (see Paragraph 3, Section 1, and Figure 2). It is not really the case of “real” against “fake” since both architectures should be valid under the same search space. Therefore, the motivation for using GAN here is a bit unjustified or unclear.

2. The paper claims that GNN2GNN is able to propose NNs from a single step and shifts the paradigm of NAS from searching to learning architecture design criteria. I would argue that this model still needs the power of searching given the presence of the valuer module, which requires the evaluation of specific architectures. For the specific experiments in this paper, the cost of searching is in the build-up of arch-performance pairs in NAS-bench101, which is further used by the model.

3. The comparison with other NAS methods is insufficient. I only see 3 baselines. Also, in table3, is “Random” stand for randomly picking architectures from the whole search space? Or do you sample a lot of architectures and take the best? If it is the case, how many samples do you use? According to [1], random graph models should perform quite well or on par with other search methods if you evaluate many samples. However, the current gap is huge, which is unconvincing to me.

4. Given the setup, the scope of this method is somehow limited to well-evaluated search spaces like NAS-bench-101 or NATS. It is unclear whether the results would generalize to more challenging scenarios, e.g., larger search spaces or more difficult tasks.

5. How many architecture samples in NAS-Bench101 are used when the generator is trained? The author mentioned removing 10% best arches. Does that mean the rest of 90% of arches are used? I hope the author can clarify this since the number of arch-performance pairs is important to evaluate how well an algorithm performs on NAS-Bench-101.

6. In table 4, the authors show how the model behaves when different numbers of architectures are removed from. It would be more interesting to see an ablation, e.g., when you randomly dropped 99% percent of the performance-arch pair in NAS-Bench101 how would the model behave. This is because even 10% of the original NAS-Bench101 is still a lot of architecture-performance pairs and we typically could not have access to that many model evaluations in NAS when model training is required during evaluation.

[1] Xie, S., Kirillov, A., Girshick, R. and He, K., 2019. Exploring randomly wired neural networks for image recognition. In Proceedings of the IEEE/CVF International Conference on Computer Vision (pp. 1284-1293).


**Q7 Justification For Your Score:**

Based on my concerns above, I do think the paper needs significant improvement before publishing.

**Q9 Complying With Reviewing Instructions:**

1: Yes.

---

### Decision · Program_Chairs · 2022-05-15

**Decision:**

Accept (Poster)

**Comment:**

Meta Review: The paper proposes a graph neural network-based framework for generating neural network architectures in an adversarial fashion. All reviewers appreciate the novelty and simplicity. Reviewers 1 & 2 suggest more comparisons with other NAS methods. In all, the meta-reviewer considers the pros to outweigh the cons, and recommends acceptance. The authors need to incorporate the reviews when preparing the camera-ready version.